# Catholicism in the Changing Religious Field of Latin America: A Mapping

**Jakob Egeris Thorsen**

Department of Theology, School of Culture and Society, Aarhus University, 8000 Aarhus, Denmark; teojet@cas.au.dk

**Abstract:** This article presents a mapping of the changing religious landscape of Latin America, specifically focusing on the place of Catholicism therein. It explores how the varying forms of Catholicism in Latin America reflect a reality of mixed modernities, described as "tiempos mixtos" (Waldo Ansaldi), where elements of pre-modern, modern, and late modern worldviews and values are intertwined in ways very different from those of the North Atlantic West. By applying the modernization and secularization theories of David Martin and Charles Taylor and the sociology of religion of Pierre Bourdieu to the Latin American context, this article takes the first step to developing an explicatory map that can help us better understand changes within the religious field in Latin America today and the role of both popular and institutional Catholicism therein.

**Keywords:** sociology of religion; religion in Latin America; religious transformation; Catholicism; Pentecostalism; conversion theory; modernization; secularization; Pierre Bourdieu; Charles Taylor

## 1. Introduction

For scholars studying religion and the Catholic Church in Latin America, the development since the 1960s has been one of changes and surprises. When living and travelling in Guatemala as a high school exchange student in the mid-1990s, I was astonished by the amount and diversity of Evangelical churches all over the country.[1] Whether in the country's capital, the indigenous highlands or the southern lowlands, thousands of small Protestant churches in interim buildings or storefronts caught my eye when crossing cities, towns, and countryside by bus or in the back of a pickup truck. Preachers would often step up into busses and deliver fiery sermons before eventually selling religious merchandise: cassettes, booklets, or cards with Bible quotes. Though Guatemala was and is a hotspot for Evangelical growth, it was not the only country where massive religious change was ongoing, and obviously, I was not the only one who had noticed. Having expected to find mainly colonial-style Catholic churches, picturesque folk religious practices, and socially engaged and liberation-oriented priests and laity, I was astonished, and I rushed to the library upon returning to home. David Stoll's (1990) book *Is Latin America Turning Protestant? The Politics of Evangelical Growth* and David Martin's (1990) *Tongues of Fire: The Explosion of Protestantism in Latin America* described and gave explanation to the phenomenon witnessed, and made it clear that the Catholic religious monopoly had long been broken and that the religious landscape was changing all over Latin America. Furthermore, it was clear that this religious transformation was intertwined with societal and political changes. A decade later, in 2005, I was studying theology and social anthropology and conducting field research for my MA thesis about the reaction of the Catholic Church in Guatemala to the massive rise of Evangelical and Pentecostal churches in the country. I wanted to explore how an once majority church had pastorally replied to being so fiercely challenged by eagerly (and sometimes aggressively) missionary Protestants, who from a Catholic perspective had 'sheep-stolen' up to one third of the flock within little more than two generations. In my research, I found that the Catholic Church—especially on the level of the laity—had turned

missionary itself, that many lay groups had adopted a Pentecostal style of prayer and praise, and that parishes were trying to educate, awaken, and prepare parishioners in order for them to be able to withstand Evangelical mission campaigns. Lay people were being formed to be ambassadors for the Catholic faith rather than passive adherents. Andrew R. Chesnut's book *Competitive Spirits: Latin America's New Religious Economy* (2003) and Steigenga and Cleary's (2007) anthology *Conversion of a Continent: Contemporary Religious Change in Latin America* helped me fathom how a general transformation of the religious field was unfolding. The change was characterized by revivalism and a general spread of Pentecostal and Charismatic spirituality, styles, and practices that affected the inner life of the Catholic Church almost just as much as its Protestant counterparts. This development was confirmed by Pew Forum studies in 2006 and 2014 which both pointed to a widespread 'Pentecostalization' of religion in Latin America, including the Catholic Church (Pew Forum 2006, 2014).[2] As I would later learn, this development was not only detectable on a local parish level, but also in the language and new pastoral strategies of national bishops' conferences and in the final document of the Fifth CELAM Conference of Latin American Bishops in Aparecida in 2007 (CELAM 2007; Arntz 2008; Libanio 2008; Thorsen 2013). Like Cleary, Chesnut, and Henri Gooren had, I therefore directed attention to the Catholic Charismatic Renewal, the largest Catholic lay movement with more than 100 million adherents in Latin America. This movement had until then to a large extent been incomprehensibly ignored by the academia (Gooren 2010a, 2012; Chesnut 2003; Cleary 2011; Thorsen 2012, 2015).

As mentioned, religious changes and overall societal change are linked to the rapid processes of modernization. In the second half of the twentieth century and first decades of the twenty-first, Latin America experienced explosive population growth, urbanization, the rise of dictatorships and armed insurgency, re-establishment of fragile democracies and a neo-liberal economic order, new patterns of work and production, economic crises and growth, and the rise of left- and right-wing populism. Each country has its own history and particularities, and Latin America is too large and complex a region to make bold and sweeping generalizations. However, despite all differences and local details, there are phenomena observed across the region that invite scholars to look for tendencies and trends which are more general, and for explanations and understandings that paint with a broad brush. This is my attempt here and it therefore goes without saying there are religious (sub-)groups—both Catholic and non-Catholic—that could rightfully have been addressed and included but are not mentioned in this article.

Though challenged by pluralism, the Catholic Church is undoubtedly still the single most important player in the religious field, and despite differences between the churches of each country and diocese, it is a hierarchical international organization, where global and regional trends manifest themselves across Latin America. This article is thus an attempt to present a mapping that captures the transformations of the religious landscape in Latin America from the mid-20th century until today, with a specific focus on the Catholicism. It is an attempt to combine intra-religious developments, e.g., the emergence of new religious groups and movements, with extra-religious changes in society, with a special emphasis on the impact of modernization. Likewise, it is an attempt to capture the interplay, competition, and mutual influence between the non-Catholic groups and the Catholic Church. The mapping combines the modernization and secularization theories of David Martin and Charles Taylor with the sociology of religion of Pierre Bourdieu, which are applied to the Latin American context in order to examine how processes of modernization affect the worldviews and practices within the religious field and Catholicism therein (Taylor 2007, p. 135; Bourdieu 1971; Martin 2005; Rey 2014).

In the following, I briefly explain the core components of the map.

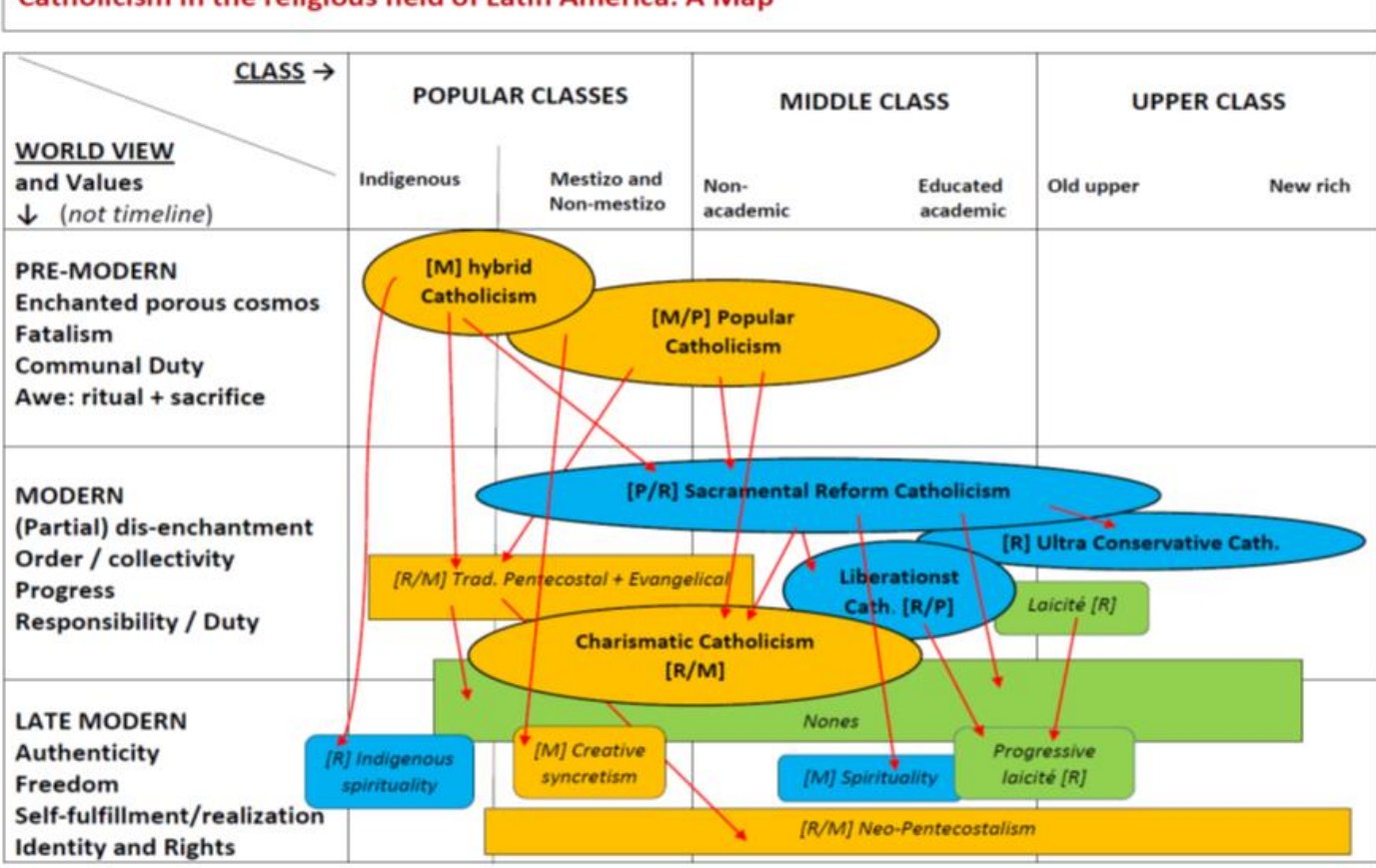

The left column indicates the different worldviews (and their characteristics and values) that coexist in Latin America, and which are here labelled as either pre-modern, modern, or late modern. The upper row indicates the different social classes found in Latin America. The space between these two axes lays out the religious field and the different religious types found there. There is a special focus on the different types of Catholicism, but all main non-Catholic types are also depicted. The colors of the figures indicate whether the worldview of the indicated form is primarily 'porous and enchanted' (orange), 'disenchanted religious' (blue), or 'disenchanted secular' (green). Furthermore, every religious type has one or two capital letters attached, indicating which of the Weberian/Bordieuan religious types dominates: the 'priestly' (P), the 'prophetic' (R), or the 'magic' (M) type. Finally, the red arrows indicate the main movements of people between the types. It goes without saying that for real people there might very well be an overlap, i.e., when people are participating in, and identifying with, more than one group and one religious type (see e.g., De Theije 1999, p. 109). First, let us very briefly present the types of Catholic ideals included in the map.

*Sacramental Reform Catholicism.* This type refers to mainline Roman Catholicism practiced in an institutional setting and centered around regular participation in the liturgy and reception of the sacraments. After the First and the Second Vatican Council, the institutional church in Latin America, weakened by reforms of independent liberal governments and by an extreme priest shortage, tried to implement orthodoxy by mobilizing and organizing the laity, e.g., in the Catholic Action movement and later in ecclesial base communities and the variety of new lay movements that spread after Vatican II. Furthermore, in the period

from 1940 to 1970, the Church received a large influx of European and North American priests and religious sisters, who helped strengthen the institution and implement reform.

*Popular Catholicism* was the most common type of Catholicism until the second half of the 20th century and the backdrop against which sacramental reform Catholicism was implemented. Popular Catholicism was and is characterized by irregular participation in institutional religion, an enchanted belief in supernatural beings, often strong relationships to specific saints, and participation in popular devotions and the *fiestas* celebrating patron saints.

*'Hybrid' Catholicism* is a type of popular Catholicism often found among indigenous peoples. It exists in a variety of hybrid forms that combine Roman Catholic and pre-Columbian beliefs and practices. Until recently this these religious forms were marginalized by the dominant groups. One could very well consider grouping some of the Afro-Caribbean and Afro-Brazilian religious types in this category.

*Charismatic Catholicism* is a very widespread type. It finds its form as a renewal movement centered around the experience of God in often ecstatic ways and the practice of the Pentecostal elements of spirit baptism, glossolalia, prophecy, and—most importantly—prayer for healing (bodily and inner). Loud, rhythmic music of praise and fiery lay preaching are key components of the movement. The Catholic Charismatic Renewal (CCR) is the largest lay movement in the Catholic Church in Latin America, but the charismatic movement has spread far beyond the original CCR organization, which today harbors only a minority of charismatic worshippers, who—depending on the country—make up between 15 and 60% of Catholics. There is thus a considerable overlap between charismatic Catholicism and sacramental Reform Catholicism since most charismatics also attend the regular Catholic Sunday mass.

*Liberationist Catholicism* is a small, but influential, progressive type of Catholicism. This wing of the Catholic Church was inspired by liberation theology in the 1970s and 1980s, when it organized the laity in ecclesial base communities in order to foster both religious and political reform. The movement influenced the Catholic educational system, and inspired grassroots organizations and NGOs. Its influence was strongest in Brazil but continues to inform Catholic engagement for social justice throughout the Americas.

*Ultra-Conservative Catholicism* is a type often found among elites. The Opus Dei is a prime example of a movement representing this type. Albeit small in numbers, it is nevertheless influential in the Church and among the middle class and elites in Latin America. Other movements, sometimes less conservative than Opus Dei, could be mentioned as representing this type, e.g., the Schoenstatt movement, Communion and Liberation, or the Neocatechumenal Way. Again, there is an overlap with sacramental Reform Catholicism, but it does make sense to single out this movement as an influential elite phenomenon.

The most important non-Catholic types are the following:

*Traditional Pentecostalism + Evangelical* is the type that represents the biggest religious challenge to Catholicism in Latin America. As we shall see below, in Latin America it makes sense to group Evangelicals and Pentecostals together, since most Evangelicals are at the same time highly Pentecostalized. Together with Neo-Pentecostalism, this category makes up around 20% of the Latin American population.

*Neo Pentecostalism* is a special type of Pentecostalism. The category covers the variety of post-denominational new, Pentecostal-style churches that have emerged throughout Latin America, including independent mega churches and corporate-like churches such as Edir Macedo's (in)famous Universal Church of the Kingdom of God. Unlike traditional Pentecostal churches, the Neo-Pentecostal churches also have a substantial following among middle-classes and new economic elites.

The *Laicité* type covers the old, liberal position of promoting secularism, especially regarding state and politics. This approach, which is often combined with a nominal Catholicism, is typically found among the old elite and the urban educated middle and upper classes.

*Progressive laicité* is the newest type, which is currently gaining influence among young, educated people from the middle classes, who demand progressive legal and cultural reforms, promoting the rights of women and minorities, e.g., by legalizing abortions and same-sex marriages, fighting violence and sexism against women, and opposing traditional gender roles. This group is often prophetically anti-religious, opposing both Catholic and Evangelical influence in society. There is a substantial overlap between this type and the 'nones'.

'*None*' is a non-religious type, representing agnostics and atheists, who do not self-identify with any religious institution or tradition. The 'none' group consists primarily of baptized Catholics, who no longer adhere to or practice their religion. Increasingly ex-Pentecostals and ex-Evangelicals identify with this type as well. In 2020, there were 4.1% 'none' in Latin America, and the number is growing.

The last three types in the map are the *Spirituality* type among the middle classes, the *Creative Syncretism* type among the popular classes, and the *Indigenous Spirituality*-type among the emerging indigenous educated middle class. The first group covers a diverse and wide array of New Age practices and beliefs, the second the new urban forms of syncretism, such as e.g., the devotion to Santa Muerte in Mexico and beyond. The last group is the emerging group of anti-syncretic neo-indigenous spirituality, which has been consciously purified of its Catholic elements.

All further elements and the theoretical rationale behind them are explained and argued for below. Before continuing unfolding the ideas behind the map, a workable definition of religion is needed that is broad and flexible enough to cover the vast religious landscape of Latin America, which addresses the highly practical character of lived religion here, and which is at the same time compatible with the work of the different thinkers applied in this study. I have chosen a working definition that combines those of the sociologist Gustavo Morello S.J., the anthropologist John P. Hawkins and the philosopher Charles Taylor. Morello defines religion as "an ongoing human relation with a suprahuman power" and Hawkins as the "system, symbols, practices, and meanings accompanying gift giving and reciprocity with some portion of the nonhuman universe" (Morello 2021, p. 36; Hawkins 2021, p. 325). Taylor, on the other hand, explicitly does not attempt to make a general definition, but provides one that captures the Western or Latin Christian understanding of religion, which in his eyes centers around a "Reading of 'religion' in terms of the distinction transcendent/immanent" (Taylor 2007, p. 15). This clear distinction cannot be made in the same way in Latin America, but Taylor adds that the purpose of religion and religious practice seem to be the double goal of ordinary human flourishing and that which goes beyond immediate human flourishing, i.e., a greater good or "a flourishing beyond" (e.g., salvation, nirvana, universal restoration or balance), the second of which might—in some cases—demand the renunciation of the former (Taylor 2007, pp. 17–19). This identification of a double purpose is worth holding on to since it captures an ambiguity found in most religions and facing religious practitioners; the ambiguity between seeking concrete, immediate flourishing or—sometimes at the cost of that—seeking a higher goal by adhering to higher principals or renouncing immediate social or material benefits. Our working definition of religion is thus: *An ongoing human relation of gift giving and reciprocity with a suprahuman power in order to pursue human flourishing in this life and ultimate goals beyond.* In this definition, the relationship and the practices (e.g., prayers, promises, offerings, sacrifices, or services) upholding that reciprocal relationship are at the center of attention. When people change their religion, they change their way of engaging with the suprahuman, whereby the character of their relationship and their understanding of it is altered, while the reciprocal core remains intact. When people drift away from religion altogether, their relationship with the suprahuman evaporates and is often exchanged for an ideal of exclusively immanent individual and collective flourishing (Taylor 2007, p. 19).

## 2. *Tiempos mixtos*: Mixed Modernities and Religious Change in Latin America

An important primary feature of the map is its attempt to capture the simultaneity of pre-modern, modern, and late modern worldviews in Latin America's religious field. Due to the region's particular history of colonialism, Creole oligarchy and liberalism, a troubled 20th century of Cold War dictatorships, and neo-liberal economic reforms, modernization in Latin America has taken a different shape than in Western Europe and the USA, which have paradigmatically defined the content and values connected with the term. This is of no surprise, since even in its Western heartlands, modernization is not in any way a straightforward and unilineal path to a society dominated by individualism, autonomy, rationalism, separation of work spheres, a distinct work ethic, and secularity. This is even less the case in class- and race-divided Latin America, where different aspects of modernization have affected sectors of society and groups of people in different ways and with varying intensity. The Argentinian sociologist Waldo Ansaldi has characterized this phenomenon as *tiempos mixtos* ('mixed times'), where modern, pre-modern, and postmodern logics, worldviews, and social practices are intertwined and overlapping, each dominating and affecting different areas of life (Ansaldi 2000). Another way of describing it is S.N. Eisenstadt's concept of 'multiple modernities' (Eisenstadt 2000). These concepts of mixed or multiple modernities are also especially helpful when analyzing the religious field in contemporary Latin America, where different religious worldviews, cultural logics, and forms of organization co-exist and dominate different parts of the religious landscape.

As in most other parts of the world outside Europe, in with regard religion is more of an exceptional rather than a paradigmatic case (Davie 2002), modernization has not automatically led to widespread secularization in Latin America (Hagopian 2009, p. 14). Latin America remains vehemently religious and if anything, the religious competition between Evangelicals and Catholics in recent times has led to an overall increase in religious practice within what Rodney Stark and B.G. Smith have termed 'the churching of Latin America' (Stark and Smith 2012). The reason for this is addressed below. However, the way that people are religious is changing: While large groups have been drawn to ecclesial forms of Catholicism and Protestantism, which emphasize adherence to orthodoxy and morality, one also sees other significant groups identifying with late modern self-expression values and hence with a form of religiosity that has looser ties to organized religion. The group of 'nones' is therefore currently growing rapidly, especially in the metropolises and among the educated middle classes (Inglehart 2009, p. 87; Loaeza 2009, p. 96; Parker Gumucio 2009, p. 131).

In order to define the categories of 'pre-modern', 'modern', and 'late modern' and to describe the characteristics of religiosity in each of them, I rely on the work of Taylor.

## 3. Charles Taylor's Notions of Mobilization and Authenticity

Taylor's (2007) opus magnum *A Secular Age* is well known across academic disciplines, and a thorough presentation of his work would therefore go beyond the scope of this article. Taylor describes how secularization became a possibility in the West and how the conditions of belief changed to the point where having and practicing a religion became one personal option among others, and where an exclusive immanent humanism would gradually become the dominant worldview. The latter is an understanding of life wherein human flourishing is the highest good that we co-operate to achieve, a good without any reference to God or a transcendent reality. The point of his book is that this process was a result of successive inner-Christian reform movements beginning in the 12th century, rather than religion simply being pushed back by enlightenment and pure reason. As Taylor himself clearly states, his is a story of the (post-)Christian West and therefore not automatically transferable to other parts of the world (Taylor 2007, p. 1). His analyses of religious change are nevertheless highly relevant in a Latin American context. Despite all differences, Latin America is an overwhelmingly Christian region that is historically closely related to Europe and North America. The strength of Taylor's approach is his interest in the general social and religious imaginaries, rather than a narrow focus on elite positions.

In this way, it is possible to understand how ideas and understandings disseminate slowly among the broader population and how they are negotiated (and sometimes opposed) in the process. What then characterizes pre-modern, modern, and late modern religiosity, according to Taylor, and how can this help us understanding the religious landscape of Latin America?

Premodern religiosity unfolds within an enchanted worldview, where the borders between the transcendent and the immanent realms are porous and hence open to influence from the spiritual world. Religious rituals help fencing of the malevolent spiritual forces and mobilization of the benevolent for protection (Taylor 2007, p. 35). In the premodern Christian era, the understanding of time was multidimensional. Profane time co-existed with 'higher time', which would break into profane time, connecting the everyday with eternity. Drawing on Mircea Eliade, Taylor adds that 'higher time' would be more than the Christian (or platonic) understanding of eternity. It also refers to the much more widespread and almost universally known 'time of origins', foundation or creation time, where people would ritually connect with the ancestors and the forces that had established order out of chaos 'in tempus illud' (Taylor 2007, p. 195). The universe was understood as an ordered cosmos, where the worldly powers (despite all their flaws) mirrored the heavenly powers, and where the rituals and sacrifices of humans helped uphold the cosmic balance and keep evil and chaos at bay. As we shall see below, Taylor's notions of enchantment and porosity are most useful when studying popular religion in Latin America, both in its traditional rural pre-modern form and its late modern urban versions.

In the 18th century, Enlightenment ideas and providential deism arose among European elites and began to challenge the unified culture of Christendom in Europe. Finally, the French Revolution cracked the sacred canopy of a well-ordered cosmos. New ideas trickled down to the popular masses and combined with the huge social transformations of the 19th century—population growth, urbanization, and industrialization—the general social imaginary changed, and so did religiosity. Taylor calls the period beginning in the early 1800s "the age of mobilization" and describes it as

> a process whereby people are persuaded, pushed, dragooned, or bullied into new forms of society, church, association. This generally means that they are induced through the actions of governments, church hierarchies, and/or other élites, not only to adopt new structures, but also to some extent to alter their social imaginaries, and sense of legitimacy, as well as their sense of what is crucially important in their lives or society (Taylor 2007, p. 445).

In Christian Europe this meant that religion became more than a given social institution that one was born into, and which was tightly connected to life in the rural village. Religion was increasingly questioned and challenged, e.g., by enlightenment reformers and the nascent labor movements, and churches therefore had to organize and rally the laity around the cause of Christianity. The disintegration of the cosmos worldview did not mean loss of a sense of order and of God as the ordering principle. It did, however, mean that the implementation of divine order in the secular sphere was the responsibility of man, organized in the church. It was a period characterized by an *esprit du corps*, where the ideal individual was seen as someone willing to sacrifice his or her work and life for the greater good of church, society, or empire, and where there was a perceived intimate connection between Christianity and the building of civilizational order (Taylor 2007, pp. 159, 430, 471). Missionary societies and apostolic orders would spread the Gospel, education, and health services ad intra to the disadvantaged and ad extra to the unbaptized peoples in the missionary territories of the colonies. The aim of the great awakenings of the 19th century was the internalization of the Christian message and the effective organization of the laity. The church would clamp down on what was perceived to be superstitious and magic aspects of folk religiosity and would promote transparent doctrine. Furthermore, in the 19th and early 20th century, the Catholic Church would try to channel the forces of popular religious fervor away from local and sometimes theologically dubious practices into church-sanctioned devotions and mobilizing events, such as pilgrimages (e.g., to

Lourdes, Fátima, Guadalupe) and Eucharistic congresses (Taylor 2007, p. 469). However, the age of mobilization (1800–1968) and of mobilized religion did not mean the complete eradication of pre-modern religious beliefs and practices, but rather that they were pushed back and increasingly associated with the unlearned rural sphere. Here they would survive far up in the 20th century, especially were modernity failed to fulfil its promise, as John Milbank points out in an insightful essay on Taylor's book (Milbank 2010, p. 60). As we shall see below, this insight is of great importance for Latin America, where—as pointed out by Gustavo Morello—the forces of modernity would leave large sections of the population excluded from its benefits (Morello 2021, p. 21).

Taylor names the latest stage of modernity in the West 'the age of authenticity' and though no period has an exact point of beginning, the year 1968 is emblematic of the time. It is characterized by increased secularization, a skepticism toward old authorities, and a quest for individual meaning-making, free choice, pluralism, and tolerance (Taylor 2007, pp. 478, 486). It is a late modern 'romantic' countermovement to the ongoing Western strife for universal order and uniformity. This period has popularized and expanded ideals of expressive individualism, authenticity, anti-authoritarianism, and nonconformity, which hitherto had mostly been cultivated in the creative upper classes, into a new mainstream culture in the West. It has challenged organized and institutionalized religion and religious organizations, almost all of which have since experienced a steady decline in membership and practice throughout the Western world (ibid., pp. 499, 510, 539). In the West, the age of authenticity and expressive individualism does not represent a rebellion against the values of the age of mobilization, such as order, progress, or universal benevolence, nor is it to be identified with sheer egoism. The project of implementing a 'modern moral order' (MMO) has not been cancelled but has transformed into an ideal of a universal order of mutual benefit, based on an exclusive humanism that potentially will enable everyone to enjoy his or her own individual freedom as long as it does not interfere with the freedom of others to "do their thing". The exclusive and secular humanist worldview that had hitherto been an elite position have become popularized and walks hand in hand with the ideal of a non-religious-based MMO of universal benevolence (Taylor 2007, p. 475). The personal and collective discipline needed for upholding the MMO has been internalized in the modern individual to a degree where it can be relaxed in some areas of life, e.g., in leisure time, the sexual domain, etc.

## 4. Religious Mobilization and the Conditions of Religious Belief and Practice in Latin America

In Latin America, this latter development is also found among the upper classes and in the educated upper middle class, but not so much in the popular classes. Taylor and David Martin make the point that the situation is very different for the vast majority of Latin Americans, who from the second half of the 20th century and onward transitioned directly from a rural pre-modern subsistence economy to an urban (post-)modern service economy in societies almost without any social security (Martin 2005, p. 31). The parallel development of the religious field carries many of the characteristics of the 'age of mobilization' in the West: religious awakening and a focus on disciplining daily life in order to avoid the dangers of the modern city (e.g., alcoholism, unemployment, social ills), holding together the nuclear family, and being part of a reliable social network. The massive rise of Pentecostalism and Evangelicalism in Latin America is a prime example thereof, but the re-organization of the laity in 'base communities' and parish-based prayer groups within the Catholic Church and the rise of the Catholic Charismatic Renewal and other lay movements show that institutional Catholicism has very much adapted to the mobilization mode (Taylor 2007, pp. 452–55; Stark and Smith 2012). According to Martin, the reason why Latin America experienced a rise in organized religious practice rather than a European-style secularization in the late 20th century is that the mixture of Latin European anti-clericalism and Anglo-American pragmatism held by the Latin American elites never spread downward to the broader population, which until the 1950s mostly

adhered to a popular Catholicism with a weak institutional anchoring and a strong pre-Columbian influence. In a continent with a "layer of Spiritism", where the porosity and enchantment of pre-modernity did not evaporate with the breakdown of traditional society, Pentecostalism has been a perfect vehicle of entry into the modern world, since it takes seriously the spiritual realm while at the same time providing the disciplinary tools to handle both the mundane and the otherworldly realm (Martin 2005, pp. 31, 71–73).

The fact that the ethos and social imaginary of Taylor's 'age of mobilization' seem to dominate the broader masses and their religious preferences in Latin America does not mean that values associated with the 'age of authenticity' are absent. Especially within the urban and educated middle classes, many have a religious practice (or not) and hold beliefs (or not) that mirror self-expression values and a detachment from any kind of moral or dogmatic rigorism (Castro et al. 2020; Pew Forum 2014, p. 10). During the last decade there has been a sharp decline in people identifying as Catholic, especially in Chile, Argentina, and Uruguay (Latinobarómetro 2017, 2021). Most probably, this owes partly to the sexual abuse scandals involving members of the Catholic clergy and partly to the often heated public debates about same-sex marriage and legal abortion in many countries, where student and women's groups have successfully mobilized against traditional Catholic positions (Taylor 2022; Artazo et al. 2021). These developments show that Catholicism in Latin America may in the future face a decline akin to that in post-Franco Spain or in Ireland in the 2000s and 2010s. Young people especially opt out of regular practice and become what late Pope Benedict XVI derogatively labelled 'cafeteria Catholics', who pick and choose from the Catholic (moral) tradition, if they do not opt out of the church altogether. According to Latinobarómetro, the percentage of Latin Americans self-identifying as Catholics dropped from 69 to 59 in just three years (2014 to 2017) and in Chile the number dropped to around 50% in 2020, while 35% according to this poll now identify as 'none' (Latinobarómetro 2021).[3] According to another survey by the Pontifical Catholic University of Chile, the number of Catholics in the country is as low as 45% (compared to 70% in 2006).[4] There is a possibility that the 2010s might in hindsight be understood as a turning point, where the values of authenticity and self-expression shifted from being an elite position to becoming mainstream in larger parts of Latin America.

## 5. Latin America's Religious Field and the Place of Catholicism Herein

Modernization and the societal change following it are important drivers for religious change in Latin America, explained by Taylor's account of how the breakthrough of the 'age of mobilization' and subsequently 'the age of authenticity' has transformed the religious field in different contexts. The sociology of religion of Pierre Bourdieu, meanwhile, can help us understand the dynamics of change within the religious field. Bourdieu develops his model from Max Weber's theory about religious ideal types (the priest, the prophet, and the magician), but shifts the attention from the ideal types to the relations *between* them, a theme that Weber does not develop (Bourdieu 2006, p. 120; Weber 1976, p. 261).

In a first rudimentary and over-simplified application of Bourdieu's theory on the Latin American religious field, the Catholic Church represents 'the priest', the Pentecostal challenger 'the prophet', and Afro-Caribbean and indigenous hybrid religion 'the magician'. Phrasing it like that is already a development of the concepts, since both Weber and Bourdieu understood the priest, the prophet, and the magician as *individual* actors within the religious field. I have elsewhere argued that we need to broaden this understanding substantially and rather speak about the 'priestly', 'prophetic', and 'magic' (or 'shamanistic') threads within religions. In most contemporary religions more than one thread is present, although one tends to dominate (Thorsen 2013). Within the religious field, the different religious actors and groups accommodate to each other. In Latin America, for example, institutional Catholicism—the priestly religion par excellence—reacted to the rise of the challenging Pentecostal religion by adopting prophetic counter-cultural elements as is seen in the Catholic Charismatic Renewal. Like their Pentecostal brethren, charismatics oppose popular Catholic traditions and the permissive and carnivalesque elements of Catholic *fiesta*

culture. In the same strain I argued that the success of charismatic Christianity (whether Pentecostal or Catholic charismatic) has been its ability to combine prophetic elements with a porous, enchanted worldview and 'shamanistic' rituals of exorcism, purification, and—most importantly—healing (Thorsen 2013). By broadening the focus from the religious leader to a religious thread (or type), one meets some of the well-founded criticisms of Bourdieu, namely that he has formulated a theory of European Tridentine Catholicism, rather than of religion in general. Bourdieu regards the laity as almost completely passive in the face of a strong and dominant church institution. Danìele Hervieu-Léger and Cristián Parker have rightfully made the point that Bourdieu's model suffers from a reductionism, which does not give voice and does not acknowledge the creative religious agency of the laity. Religious people in Latin America and elsewhere are not passive consumers of religious goods in a market, they are also 'producers' of those goods themselves and may push or drift towards either the prophetic, priestly, or magic thread, whereby they create new religious institutions (e.g., Evangelical churches) or force the existing to react and adapt (Rey 2014, pp. 124–25; Parker 2015, p. 239; Thorsen 2013).

## 6. The Importance of Social Class in Latin America

Finally, before delving further into the changes described in the map, we must briefly address the upper row, indicating the social classes in Latin America. Social class and religious adherence and practice are closely connected and must therefore be considered. Across all national differences, it is possible to divide the population into three general categories: lower, middle, and upper class, which can each be subdivided further.

The lower class represents the majority population and is constituted on the one hand by urban workers and the 40–70% of urban dwellers who engage in the large informal economy. This latter group engages in irregular forms of wage labor, often part-time without the benefits and protection of a legal contract. Many are self-employed, e.g., operating unregulated micro-enterprises producing or selling goods and services from their home or in the street. Others work in low-wage domestic service to middle- and upper-class and households as maids, housekeepers, or gardeners. In the slums, some people engage in illegal activities, e.g., prostitution, contraband, or the drugs trade (Veltmeyer and Petras 2011). The other part of the lower class is the rural proletariat and peasantry: smallholders of land, (seasonal) wageworkers, who engage in subsistence farming combined with the informal economic activities listed above. In general, the lower class in Latin America is *mestizo*, i.e., of mixed descent. Furthermore, the majority of Latin America's minority population of indigenous people and African-Caribbeans belong to the lower class.

The middle class makes up between 15 and 30% of the population, depending on the country. It includes rural middle-size farmers, business owners, and public employees. This latter group includes teachers, technicians, social workers, bureaucrats, and office managers. Since the neo-liberal reforms of the 1980s, many public employees have seen a sharp decline in real wages, social status, and job security, and one can reasonably argue that many from this group can today be categorized as low class, e.g., police officers (Veltmeyer and Petras 2011). In the map, the middle class is sub-divided between non-academic and academic, to capture the transition from values of 'mobilization' to 'authenticity' found especially within the latter group. Finally, there is a small upper class, divided between an old and a new upper class representing nouveaux riches capitalists who have benefitted from neo-liberal reforms from the 1980s and onward.

## 7. Analyzing Types and Religious Change within the Map

In the last part of this article, I address the movement between main types in the map and how this can make sense of religious change in Latin America. The four religious types with the largest followings are undoubtedly mainstream sacramental Reform (SR) Catholicism, 'traditional Pentecostal and Evangelical, Neo-Pentecostalism, and charismatic Catholicism. Likewise, the non-religious type 'none' is gaining influence. The main change the last fifty years has been from 'hybrid' and 'popular Catholicism' to one of the five

mentioned groups. Furthermore, there has also been some movement—though to a lesser degree—from SR Catholicism to (Neo-)Pentecostalism, charismatic Catholicism, and to 'none'. According to the Pew Forum, in 2014, 69% of Latin Americans were Catholic and around 19% Protestant (Pew Forum 2014, p. 4). As we saw above, the number of self-identified Catholics is even lower in more recent polls (Latinobarómetro 2021).[5]

a.   *Catholics*

The statistics do not distinguish between popular, SR, and charismatic Catholicism since all three categories are comprised of baptized Catholics, who either regularly or occasionally attend the same Sunday mass. However, as I shall argue in a moment, it does make sense to distinguish between the three, since there are considerable differences in practice and worldview. The percentage of Catholics who identify as 'charismatics' ranges between 20% in Argentina and 58% in Brazil (Pew Forum 2014, p. 64).[6] The numbers of hybrid or popular Catholics are likewise difficult to establish, but if one assumes 'regular weekly church attendance' to be a sign of people possibly being either SR or charismatic Catholics, it is possible to associate a portion of the remainder with either (porous enchanted) popular Catholicism or dis-enchanted secularized 'nones', who might nevertheless still figure in church statistics due to their baptism. Weekly mass attendance among Catholics varies from 72% in Guatemala to 9% in Uruguay, with the largest Latin American countries ranging in between: Brazil 37%, Mexico 44%, Colombia 49%, Peru 30%, Argentina 15%, Chile 14%, Venezuela 17%, and Bolivia 35% (Pew Forum 2014, p. 43). It is difficult to tell who among the Catholics who do not attend weekly services adhere to popular Catholicism or are non-practicing SR Catholics. The statistics of 'importance of religion' and 'daily prayer outside religious services' can give us an indication, e.g., comparing Brazil, Mexico, and Colombia with Argentina, Uruguay, and Chile. In the first three countries, weekly church attendance for Catholics was 37, 44, and 49%, as we saw above. The figures for 'importance of religion' (being important or very important) among Catholics in the three countries are 71, 45 and 78% and for 'daily prayer' are 59, 40, and 73%. In the southern cone countries, weekly mass attendance for Argentina, Uruguay, and Chile was 15, 9, and 14%. Here, the Catholic numbers for importance of religion are 43, 31, and 40% and for daily prayer 38, 33, and 39% (Pew Forum 2014, pp. 41–44). These numbers give us an—admittedly weak—indication that porous-enchanted popular Catholicism accounts for a larger proportion of baptized Catholics in the first three countries compared with the last three, and vice versa regarding the more secular non-practicing Catholicism and the self-identifying 'none' type, where we must expect (ex-)Catholics in the last three countries to be present in higher proportions.

b.   *Protestants*

Regarding the categories 'traditional Pentecostal and Evangelical' and 'Neo-Pentecostal' it is necessary to briefly unfold the character of Protestantism in Latin America: It is overwhelmingly Pentecostal and charismatic. Regarding terminology, most Protestants (whether Pentecostal, Evangelical, or historical mainline) identify as *evangélicos*. According to the Pew Report mentioned above, 65% of all Protestants in Latin America either belong to a Pentecostal denomination or self-identify as Pentecostal (Pew Forum 2014, p. 8). The remaining 35% have often acquired some charismatic elements, many having added a *renovado* ('renewed') to their denominational name. In a 2006 global report on Pentecostalism, the Pew Forum focused on Brazil, Chile, and Guatemala in Latin America. In the first two countries, 78% of Protestants were 'renewalist' (i.e., either Pentecostal or charismatic), as were 85% of Guatemalan Protestants (Pew Forum 2006, p. 4). Since Pentecostalization has continued since then, I think it is fair to assert that 80–90% of all Protestants in Latin America today are Pentecostal or charismatic. The difference between traditional Pentecostals and Neo-Pentecostals is often fluid. Classical Pentecostals belong to denominations that trace their origin to the 1906 Azusa Street Revival in Los Angeles (California), and they mostly have clear doctrine about Spirit baptism, glossolalia, and other Pentecostal elements. Traditional Pentecostals have an otherworldly orientation, and adhere to modesty and

strict moral rules. They typically belong to the popular classes and the lower middle class. Neo-Pentecostalism is typically more oriented towards this-sided flourishing. It is doctrinally more flexible, and the churches are often independent and led by charismatic pastors, such as Edir Macedo in Brazil or Carlos 'Cash' Luna in Guatemala. Contrary to traditional Pentecostals, these new churches often adhere to prosperity theology, are morally less strict, promote upward social mobility and tend to engage more pro-actively in political life. Neo-Pentecostals draw comparatively more adherents from the middle class than traditional Pentecostals (Allan 2010; Doran and Garrard 2020; Garrard-Burnett 2012).

*c.     Understanding Religious Mobilization and Change*

So why have people drifted heavily from hybrid and popular Catholicism to SR Catholicism, (Neo-)Pentecostalism, and charismatic Catholicism during the second half of the 20th century? And why do we now see a new move towards non-practice as either non-practicing Catholics, 'nones', or adherents of 'progressive laïcité'? As mentioned, religious transformation is intimately linked with modernization and social change. Due to the extreme institutional weakness of the Catholic Church until the mid-twentieth century, Latin America was a place of non-institutional popular and hybrid Catholicism embedded in local—mostly rural—social structures of a population that for the majority was engaged in subsistence farming, skilled craftmanship, or local trade. Life was poor, but mostly predictable. An all-encompassing religious sense of the world (porous enchantment) was a given, and religious obligations (*cargos*—both ritual and economic) connected with the *fiestas* of patron saints, processions, and *posadas* had to be accepted, and would—if duly honored—secure a man's or a family's social position in the local community. In Bordieuan terms, popular and hybrid Catholicism had both a 'priestly' and a 'magic' dimension. Since the institutional church was most often only scarcely present, the collective public rites carried out by religious fraternities (*cofradías*) and the basic religious instruction provided by sacristans and later catechists represented the priestly dimension of local religious life. The 'magic' side of popular and hybrid Catholicism had then and has now a more individual side to it, where special saint-devotees, healers (*curanderos*), and shamans offer their services to people in need of healing, purification, or liberation from curses or spells cast on them by others (*brujería*). Hugo Suárez has termed these religious providers 'para-ecclesiastical agents', and as we shall see later, this term also well applies to the many lay preachers and healers that have emerged in connection with the Catholic Charismatic Renewal and other semi-autonomous lay initiatives (Suárez 2021, pp. 275–76).

From the mid-twentieth century, explosive population growth undermined subsistence farming as the basic and sufficient mode of livelihood and created large-scale migration to the urban centers, where low paid jobs in service, industry, or petty trade in the informal economy were an unpredictable way of survival in an often hostile social environment of urban poverty, crime, and alcoholism, lacking the established social structures of traditional society. For many of those most affected, e.g., the landless and the migrants to the cities, the reciprocal relation to God and spirits within traditional popular and hybrid Catholicism would no longer be deemed helpful in coping with the challenges of the new situation—a cultural and existential *anomie*, as described in convincing detail by John P. Hawkins for the Guatemalan case (Hawkins 2021).

The religious response to this rapid and often chaotic process of modernization can best be described as a large-scale religious mobilization of people: a churching of popular religiosity. The global Catholic Church began an extensive revival and reform program within the helplessly under-staffed Church in Latin America in the first half of the 20th century. Thousands of North American and European priests and religious sisters were sent southwards and a large-scale organization and education of the laity within the framework of Catholic Action was set in motion, promoting orthodox Catholicism centered around liturgy, sacraments, and catechism (Hernández Sandoval 2016). This strengthening of official 'priestly' religion would in many places have a 'prophetic' character, when foreign priests and young, trained catechists challenged the local religious *costumbre* ('custom'). It created inter-generational conflicts with the pillars of popular and hybrid Catholicism

when they were trying to enforce Catholic orthodoxy (Hughes 2016, p. 488). After the Second Vatican Council, mobilization continued both along the 'sacramentalist' lines just mentioned, and also within the framework of the emerging progressive liberation-oriented wing, and the new—often more conservative—lay movements such as the Charismatic Renewal, *Cursillos de Cristiandad*, Legion of Mary, Legionaries of Christ, the Neo-Catechumate Way, and others (Thorsen 2016; Vásquez and Peterson 2016).[7] Another small, but influential mobilization movement is Opus Dei, which counts many of its members among the elites and upper middle class. Though surrounded by secrecy, the influence of the movement in universities and seminaries is widespread (Ávila García 2020; Benavides 1997, p. 539; Lehmann 2016, p. 748).

The Catholic Church was not the only one mobilizing; so were the Protestant churches, both the traditional Evangelical and the newer Pentecostal, which likewise sent missionaries to establish churches throughout Latin America. The mobilizing Catholic Church and its Protestant counterparts would fish in the same abundant pool of un-catechized popular Catholics. Contrary to the hierarchical Catholic Church with its bureaucratic structures and highly educated, but scarcely available priests, Evangelical churches can be started anywhere by anyone who has some convincing preaching ability, giving the Protestants a huge competitive advantage in a time of cultural and religious break-up and rapid new mobilization. Protestants furthermore benefit from having the same social and cultural backgrounds as the people they seek to attract, whereas most Catholic priests have been shaped by years of theological and ecclesial formation, which may alienate them from the daily lives of the popular classes. Here, charismatic Catholicism provides an alternative to institutional Catholicism, since it may mirror the local bottom-up mobilization of the Protestants. As described elsewhere, the charismatic movement within the Catholic Church has in many places created parallel ecclesial structures with Charismatic assembly halls, services, and lay preachers and healers attending to the renewed Catholic populace in their area, often despite a lack of wholehearted episcopal backing. These structures often function with a very high degree of independence and can easily disconnect from the official Catholic institution if faced with a hostile parish priest or bishop (Thorsen 2015, pp. 38, 175). Catholic lay preachers can sometimes generate extra income by visiting different charismatic prayer groups, and they may very well be the most numerous and influential para-ecclesiastical agents in Latin America today. Curiously enough—as pointed out by Gooren—the official charismatic flagship organized in the 'Catholic Charismatic Renewal' (CCR) seem to stagnate in many Latin American countries despite the general trend towards Pentecostal and charismatic worship. The stagnation documented by Gooren might indicate that when the movement was institutionalized and instrumentalized by the bishops (top-down), it lost some of its spontaneity and ability to attract followers (Gooren 2012, p. 293).

*d.    Explaining Pentecostal and Charismatic Success*

What is the formula behind the immense success of Pentecostalism and charismatic Catholicism? For John P. Hawkins, Pentecostalism (and its Catholic charismatic counterpart) enables people to express a cry of angst in a culturally collapsing world. At the same time, it provides them with the tools to manage the uncertainty of the new condition: a close-knit religious community with high moral standards and mutual support, which has a strong focus on physical and emotional healing through prayer and ecstasy (Hawkins 2021, pp. 11, 291). These are the tools that help people fight what Chesnut calls the pathogens of poverty (Chesnut 2003, p. 14). The religious worldview of the new church communities remains porous and enchanted, but whereas popular and hybrid Catholicism tended to foster an attitude of passive resignation towards the chaotic forces coming from outside, Pentecostalism encourages agency aimed at individual and communal reform and adaption. Theorizing with Taylor, one could say that the strength of Pentecostalism lies in its mobilizing character, which contrary to its 'buffering' and disenchanting European counterpart, upholds a porous-enchanted worldview. This porosity is, nevertheless, perceived differently in Pentecostalism than in the former popular and hybrid Catholicism: It

is a re-modelled enchanted worldview, which has lost some of its nebulous opaqueness and has become more transparent to the faithful believer, who now manages the skills and techniques to navigate in often troubled spiritual waters, warding off the forces of evil through a strong personal relationship with the triune Christian God, known through the Son and manifestly experienced in the Holy Spirit. Both Pentecostalism and Catholic charismatism are 'prophetic' types of religion, which challenge the status quo and opt for a walk-out from the established socio-religious norms. Popular and hybrid Catholicism are the target of both, who attack the permissive attitudes to alcohol, many unorthodox practices, and the festive and carnivalesque elements, which seem to be the complete opposite to the disciplined ordered life pursued by the renewalists. Pentecostals furthermore understand themselves over and against the Catholic Church, whom they hold accountable for the unorthodox character of popular religion and the lax attitude towards churched religion. Furthermore, they launch the traditional Protestant critique of Catholic faith and practices (worship of saints, ordained celibate priesthood, Eucharistic sacrifice, etc.), which is presented with an un-ecumenical eagerness and self-confidence, in a few extreme cases even leading to excesses, such as public bashing of saint statues (The Economist 1995). Charismatic Catholics, on the other hand, have a more mixed relationship with their church and fellow Catholics. Being 'prophetic', they challenge the popular religious culture, historically influenced by Catholicism, but at the same time they propose a renewed and dedicated Catholicism as the alternative. They thus conceive of themselves as true, dedicated and avantgarde Catholics, and they regard their lukewarm and lax fellow *católicos* as tainting the image of Catholicism as the true Church. Therefore, they have wholeheartedly embraced and engaged themselves in the New Evangelization and the Continental Mission campaigns launched in Latin America in the 1990s and 2000s by the Conference of Latin American Bishops (Vásquez and Peterson 2016, p. 390; Lindhardt and Thorsen 2015, p. 184). The complex relationship between the charismatics and the Church hierarchy is a study of gradual mutual adaptation between a 'prophetic' renewal movement and a 'priestly' hierarchical institution (Alberoni 1984, p. 286; Thorsen 2015, p. 69). The movement has accepted some measure of guidance and control by the hierarchy, whereas the institution has accepted a high degree of independence for the charismatics in exchange for the latter channeling their vitalizing energy into supporting the established church.

In the above, the 'prophetic' walk-out character of Pentecostalism and charismatic Catholicism has been highlighted, but following Hawkins' explanation, there is also a deep continuity with the porous enchanted 'magic' type of popular religion, stressing individual healing and liberation from spells and evil spirits. It is exactly this combination of 'prophetic' and 'magic' elements, using the Bordieuan terms, which explains the success formula of Pentecostal and charismatic religious groups in Latin America. In the midst of social turmoil and change, a pathway to modernity is offered, which at the same time allows for a deep continuity with a popular religious worldview (Hawkins 2021, p. 310; Thorsen 2013, p. 49).

*e.    Late Modern Trends*

In the above, the most important religious groups in the map have been presented, including the dominant forms of Catholicism. In the final section of this article, I will briefly present some of the new religious or spiritual groupings, which often but not always find adherents among the middle class, and whose more ephemeral and unbounded character make them less easy to delimit since they may very well be comprised of e.g., non-practicing Catholics. These forms of religion and spirituality are found in the lower part of the map, exhibiting the characteristics of late modernity, i.e., a focus on subjectivity, authenticity, and self-fulfillment and a rejection of the core values of mobilization: submission and conformity to the roles, rules, duties, and expectations of family, religious group, and wider society. In *The Spiritual Revolution*, Linda Woodhead, Paul Heelas, and colleagues describe the former as a period where "subjective life forms of the sacred, which emphasize inner sources of significance and authority and the cultivation or sacralization of unique subjective-lives, are most likely to be growing" (Heelas et al. 2021, pp. 3–4). Taylor writes that "much of

the spirituality we call 'New Age' is informed by a humanism which is inspired by the Romantic critique of the modern disciplined, instrumental agent" (Taylor 2007, p. 510). As recently documented, one finds the same variety of alternative or New Age spiritualities in Latin America as are known elsewhere (Gooren 2019). Whereas mobilizing religion appeals to those who are in need of a strong guiding community while in transition into an ambiguous modernity of risks and opportunities, New Age spiritualities seem to appeal more to people of the educated middle class, who reject what is perceived as the stifling moralism and inflexible dogmatics of churched Christianity, whether Catholic or Protestant.

Another interesting development is how once hybrid religious forms such as e.g., Maya or Andean indigenous traditionalism, which mixed Catholic and pre-Columbian elements and were practiced semi-clandestinely alongside official Catholicism, have begun a process of consciously becoming independent, often purifying themselves from Christian and Catholic elements. Sometimes this process of religious re-invention is carried through by the most educated among the groups of practitioners and is closely linked with a social and political mobilization around protecting indigenous rights and identity. In Bordieuan terms, the eminently 'magic' religious form of hybrid popular religion is thereby transformed into a 'prophetic' type, which challenges the status quo of disadvantaged minority groups and offers a walk-out from the perceived persistent suppressing colonial structures within the religious realm. These new anti-syncretistic religious forms have hitherto mostly been limited to groups within the emerging educated elite among minority populations, whereas the majority of indigenous people opt for a mobilizing religion—Pentecostal or Catholic (charismatic)—or remain attached to hybrid traditional practices. To exemplify this, I limit the discussion to a single case, namely C. James Mackenzie's description and analysis of the emergence of a purified pan-Maya spirituality in Guatemala, which he sees as an illustrative case of a religious re-creation in the light of a late modern worldview:

> First, in ideological terms sacerdotes mayas [purist Maya priests] seek to both heal the rupture caused by modernity's caesural need through connection with a re-constituted ancient Maya practice, and create a new and liberating break through the rejection of centuries of [ ... ] exposure to Christianity [ ... ]. Second, in practical terms, a modern telos of progress and freedom is seen in various works of purification and translation [ ... ], and especially the assertion of authority to engage in this work. Finally, in subjective terms, and with regard to universalism, sacerdotes mayas represent their religion as a unique cultural achievement which can be placed alongside other religious expressions of humanity, part of a multi-cultural world inhabited by a single otherwise undifferentiated human species (MacKenzie 2009, p. 371)

By developing a consciously anti-syncretistic spirituality and ritual practice, indigenous Maya reinforce their cultural and ethnic identity while rejecting its subordinate status in the racist social fabric of (post-)colonial society, with its heavy Latin and Catholic heritage. Likewise, in Afro-Brazilian and Afro-Caribbean religious traditions such as Candomblé, Santeria, or Vodou there has been an anti-syncretistic movement of 'Africanization' (Palmié 2002; Capone 2010).

Finally, one should mention the new and fast-growing examples of urban popular syncretism, such as the veneration of Santa Muerte in Mexico and elsewhere and the popularity of so-called 'narco-saints'. Contrary to many of the other groups mentioned above, these latter traditions find a large part of their following among the lower economic strata of society (Kingsbury and Chesnut 2021; Michel and Park 2014; Chesnut 2012).

## 8. Conclusions

In this article, I have presented a mapping of the changing religious field in Latin America with a special focus on the place of Catholicism therein. The map shows how societal and religious change are intimately linked and how the rapid processes of modernization and urbanization have contributed to a thorough transformation of the religious landscape. The Catholic Church and Catholics throughout Latin America have reacted to

these changes in different ways and have adapted their methods of religious organization and practice in order to meet the needs of the new situation. In the analysis above, religion was defined as *an ongoing human relation of gift giving and reciprocity with a suprahuman power in order to pursue human flourishing in this life and ultimate goals beyond*. People shift their religious practice when their traditional ways of engaging reciprocally with God (or the gods, and/or spirits) no longer seem to promise human flourishing or to point them credibly towards a higher goal. Shifting one's religious affiliation or group becomes a possibility when someone trustworthy—most often a family member, colleague, friend, or neighbor—can point out a new and more promising way of relating to the godhead(s) and, importantly, when that person or his or her religious group actively places the potential convert in a new reciprocal social and religious relationship by providing the healing, comfort, help or attention needed: Most conversions in Latin America happen in moments of crisis (Hawkins 2021, p. 321; Gooren 2005). Since the mid-twentieth century, conversions have primarily gone from popular Catholicism to mobilization-type religious groups, be it sacramental Reform Catholicism, Pentecostalism, or Catholic charismatism. The map also shows that the modernization process is not straightforward and universal, but 'mixed', with sectors of the population upholding porous enchanted worldviews with pre-modern characteristics, while others adhere to late modern values of authenticity. The *tiempos mixtos* are thus intact. Mobilization movements of (Neo-)Pentecostals and Catholic charismatics are successful because they offer a pathway to modernity while retaining the porous enchanted worldview of popular religion in a refurnished and more transparent version. Although the Catholic Church has experienced severe losses, it has done its best to mobilize, both by implementing its own reform programs of catechesis and New Evangelization and by allowing the more radically 'prophetic' (and 'magic') Charismatic Renewal to unfold and grow within its midst. The later has probably curbed the growth of Pentecostalism, especially among Catholics with some attachment to the institutional church. Although at a much slower pace than 20 or 30 years ago, the general churching and 'Pentecostalization' of religious life still seems to continue among sections of the popular classes. At the same time, we see a possible mass exodus from mobilization-mode religion among the educated urban middle class, as the declining number of self-identifying Catholics indicates. As in Europe and the USA, it appears that the Catholic Church in Latin America has a hard time adapting to the progressive and individual-oriented values of the age of authenticity, especially because its ecclesial apparatus is still geared for mobilization. The World Christian Encyclopedia predicts a growth of 'nones' from 4.1% in 2020 to 6.1% in 2050. In the light of the change towards authenticity values, this predicted growth may very well turn out to have been estimated far too low (Johnson and Zurlo 2020).

## 9. Future Research

While the mapping presented in this article can hopefully help us to better understand what produces religious change in Latin America, there are gaps in in our knowledge about the topic that merit the attention of future research, and which must be addressed before wrapping up this conclusion. As we have seen above, quantitative studies show a Latin American religious field in motion and transformation. Statistics vary considerably depending on their design, the questions used, etc., but nevertheless portray a gradual change towards pluralism, which appears stable: Sections of the population change their religious belonging and seem to remain within their new category. Behind the numbers, on an empirical level, we find a much messier picture of highly individualized stories of conversion and religious changes (sometimes multiple!), altered religious activity, and quiet disaffiliation. Already in 2004, Edward Cleary characterized it as a situation of "Shopping Around" and Henri Gooren stressed the need for studying 'conversion careers', likewise arguing that people in Latin America often experience various changes and periods of highly varying intensity in their religious lives (Cleary 2004; Gooren 2007, 2010b). While the reasons for converting to e.g., Pentecostalism are well studied, we know much less about the reasons for religious disaffiliation and passivity. This creates a dilemma: Most

researchers know about the phenomenon from field experience, but find it difficult to study people who do not practice, are disaffiliated, or are in a long period of voluntary passivity. Those people are very hard to study qualitatively for the simple reason that they do not gather for worship and often keep quiet about their (non-)practice. As Martin Lindhardt has pointed out for Chile, many third generation Pentecostals have prolonged periods without attending any church, and many have a hard time identifying themselves as clearcut Evangelicals, which leads Lindhardt to suggest that national polls should include the category *'Evangélico a mi manera'* ('Evangelical in my own way'), mirroring a parallel category that exists for Catholics (Lindhardt 2022). Likewise, within the Catholic field, we need to better understand the 'conversion careers within' Catholicism. We know why people join the charismatic groups or ecclesial base communities, but much less about why people quit them again and return to more lukewarm occasional participation in mass or stop practicing altogether. In an ever more pluralistic religious landscape, where the late modern values of authenticity become gradually more dominant among larger sections of the population, future research should address these lacunae.

**Funding:** This research received no external funding.

**Institutional Review Board Statement:** Not applicable.

**Informed Consent Statement:** Not applicable.

**Acknowledgments:** I would like to thank both of my anonymous reviewers for excellent and constructive feedback.

**Conflicts of Interest:** The author declares no conflict of interest.

## Notes

[1] The term 'Evangelical' (*evangélico*) is what members of the non-Catholic Christian churches (different traditions of Protestantism) in Latin America generally call themselves. Around 70–90% of Evangelicals in Latin America are Pentecostal or Charismatic, according to Pew Forum (2014).

[2] In this article, I rely primarily on The Pew Forum Report from 2014, since the 2006-report has a confusing categorization of Pentecostals, Neo-Pentecostals and Catholic Charismatics into one category termed 'Renewalists' (Pew Forum 2006, 2014). Scholars such as John P. Hawkins, however, made a point out of creating a joint category (in his case 'Christian Pentecostalism'), which consciously blurs the distinction between Catholics and Protestants while emphasizing the common Pentecostal/Charismatic elements, Hawkins (2021).

[3] According to the World Christian Encyclopedia, 'only' 11.1% of Chileans identified as Non-religious in 2020, which demonstrates how statistics vary in Latin America, depending on how people are asked, see: Johnson and Zurlo (2020).

[4] Cf. https://catholicherald.co.uk/chile-a-wake-up-call-for-the-global-church/ (accessed on 16 August 2022) and https://www.trtworld.com/magazine/is-catholicism-in-latin-america-set-to-become-a-minority-faith-53587#:~:text=In%20places%20like%20Chile%2C%20trust,70%20percent%2015%20years%20ago (accessed on 16 August 2022).

[5] See: https://analankes.medium.com/in-latin-america-the-catholic-flock-is-shrinking-e86e073dd7a2 (accessed on 12 December 2022).

[6] People who self-identify as Charismatic adhere to classical Pentecostal/Charismatic practices such as Baptism in the Spirit, Prayer for healing, and glossolalia. The numbers for other countries are Mexico (27%), Colombia (24%), Peru (32%), Venezuela (32%), Chile (23%), Ecuador (40%) and Guatemala (38%) (ibid.).

[7] Since the the Catholic Charismatic Renewal very large movement comprised by people from all social classes is impossible to categorize as unequivocally 'conservative'. In general, however, Charismatic Catholicism is socially conservative in moral questions and in questions about dogma. In secular politics, on the other hand, it seems to be impossible to associate Catholic Charismatics with any particular ideology.

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
