# Peer review of "Catholicism in the Changing Religious Field of Latin America: A Mapping"

_religions, doi:10.3390/rel14040461_

Round 1

Reviewer 1 Report

This reviewer has done research in Guatemala and is familiar with the data presented this paper and some of the  sources used in it. This article draws advantageously  on Taylor’s theory of historical development and Bourdieu’s sociological theories. It concentrates on Catholicism but gives an broad overview of Latin American religious change. Well done.

Author Response

Dear Reviewer 1,

thank you for your generous review!  Very encouraging.

Reviewer 2 asked for modest changes and clarifications, which have now been completed. A revised manuscript will be uploaded later today/tomorrow.

Reviewer 2 Report

The purpose of the article is to present a new model that describes the changing religious landscape in Latin America and the place of Catholicism herein (Abstract).  It delineates 7 forms of Catholicism (Sacramental Reform, Popular, Hybrid, CCR, etc.; 3-4), but only 2 forms of Pentecostalism (Traditional Pentecostalism+Evangelical[ism] and Neo-Pentecostalism) plus Laicité (2 types), and middle-class Spirituality, Creative Syncretism, and Indigenous Spirituality (4-5).  Using Ansaldi’s concept of tiempos mixtos, the author traces premodern, modern, and late-modern worldviews in these 14 religious forms.  The main factors underlying the model are modernization (including secularization theories by David Martin and Charles Taylor) and Bourdieu’s distinction between magic[al], prophetic, and priestly types of religion.  The full model is presented on p. 3 in a sophisticated color scheme.

The model aims to capture the simultaneity of premodern, modern, and late-modern elements in the Latin American religious field.  Modernization is recognized as not being a “straightforward” process (not even in the West) and defined as “a society dominated by individualism, autonomy, rationalism, separation of work spheres, a distinct work ethic, and secularism” (6).  Premodern religiosity is described as “an enchanted worldview, where the boarders between the transcendent and the immanent realms are porous and hence open to influence from the spiritual world” (7, 274-276).  Following Taylor (2007: 478, 486), the late-modern era started in 1968 and is defined by “increased secularization, a skepticism toward old authorities, and a quest for individual meaning-making, free choice, pluralism, and tolerance” (8, 335-337).

All of the above constitutes a very useful overview and synthesis of existing concepts and theories.  However, how much of this is really new?

The bibliography is excellent (although it incorrect lists Anderson as “Allan”), up to date, and relevant.  Ansaldi, Hawkins, Martin, Taylor, and other theorists are used effectively. 

However, these are my main concerns and criticisms:

1.  I think the author should present the 7 forms of Catholicism more explicitly as ideal types (3-4).  Plenty of research (de Theije, Gooren) shows that the borders between Liberationalist Catholicism and the CCR are porous; many people are active in both groups and see no contradiction in this at all.  Similarly, Popular and Hybrid Catholicism strongly overlap.  And why is Opus Dei mentioned as a separate category, but not Schoenstatt?

2.  I do not understand why Nominal Catholics are mostly listed in the model (3) under “upper class.”  There are many nominal Catholics in the popular and middle classes, too!

3.  Please add data to support the claim that Nominal Catholics are “growing” (4, line 147).  If this claim is correct it needs to be analyzed and explained.  These people still self-identify as Catholics, did not become Protestants, and did not drop out of religion altogether either.

4.  It is a strange sweeping generalization to consider the whole RCC as “priestly,” all of Pentecostalism as “prophetic”, and as indigenous/Afro-Caribbean forms as “magical” in Bourdieu’s theory (9, 415-418).  Would it not make more sense to say that the Catholic Church has priestly (Sacramental Reform), prophetic (Liberationalist), and magical (CCR, Popular, Hybrid) forms?  Likewise, one could consider the historical Protestant churches as “priestly,” neo-Pentecostalism as “prophetic,” and (classical) Pentecostalism as “magical.”

5.  The 2006 Pew report has been severely criticized for its error in conflating charismatic Catholics, Pentecostals, and neo-Pentecostals all in the same category as “renewalists” (12, line 525-530).  The 2006 report is problematic to use as a source, especially for percentages.

6.  I am puzzled why the CCR is listed together with “more conservative” lay movements such as the Cursillos, Legion of Mary, and Legionaries of Christ (13, 592-594).  What are the reasons for that?  The CCR is not political in purpose or practice; its leaders can be more progressive or more conservative, depending on their education, class, and home country.

7.  I do not agree with the characterization of the CCR as “bottom-up” (13, 611-612).  Cleary documents how various priests, bishops, and cardinal Suenens were all instrumental in supporting and spreading the CCR throughout Latin America.  After 2000-2002, the CCR was put under central clerical control, mass healings and exorcisms were cancelled, and the emphasis was shifted to lay leadership training, parish prayer groups and providing pastoral care.  This was part of a coordinated international campaign by the Vatican that severely curtailed the CCR and strongly decreased its affiliation in most countries.

8.  I think the author underestimates how far secularization has already proceeded in Latin America (just like Morello did).  Just look at the nonreligious populations of all Latin American countries in the 2020 World Christian Encyclopedia!  Guatemala has the smallest (1.4%; although Pew listed 6% in 2014), but most countries are already between 5 and 10%.  (Apart from outliers such as Cuba and Uruguay.)  Moreover, these nonreligious populations are growing strongly, especially among the younger generations (suggesting further future growth as well).  The author should explicitly mention and analyze this, especially since it fits so well with this model/approach!

9.  I think the author should explicitly address current gaps in our knowledge about (changes in) the religious landscape of Latin America and offer some concrete suggestions for future research.

I am sympathetic to the author’s ambitious effort, but I am not convinced (yet) that this effort truly constitutes a new model, i.e. that it develops a new theory that is able to not only explain what is currently happening in Latin America but also to predict future developments.  I would challenge the author to explicitly state what is new about this approach and engage in some predictions for the future.  I recognize that this is a high bar to meet; a simpler solution would be to use the word approach rather than model. 

The English is very good, with only a small number of typos and grammatical errors.  I found the word choice “antidote” (14, line 645) confusing; I assume that the author really means to say “opposite” or “antithesis”?

If the author can address all the 9 issues I raised here in a satisfying manner I believe this article will constitute an important contribution that should definitely be published.

Author Response

Dear Reviewer 2,

thank you for your thorough review and most helpful comments and suggestions.

I have responded to your questions and concerns below.

  1. I think the author should present the 7 forms of Catholicism more explicitly as ideal types (3-4).  Plenty of research (de Theije, Gooren) shows that the borders between Liberationalist Catholicism and the CCR are porous; many people are active in both groups and see no contradiction in this at all.  Similarly, Popular and Hybrid Catholicism strongly overlap.  And why is Opus Dei mentioned as a separate category, but not Schoenstatt?

                      I have now made explicit that the forms of religion are ideal types with porous boundaries. The Opus Dei type has been changed to ”Ultra Conservative Cath.” and now includes other highly mobilizing movements.

  1. I do not understand why Nominal Catholics are mostly listed in the model (3) under “upper class.”  There are many nominal Catholics in the popular and middle classes, too!

                      The error has been corrected. Due to the review, I have changed the category from “nominal Catholics” to “Nones” and expanded it to all social classes. Since there exist “nominals” within all Catholic types, the category does not work as an independent category and has therefor been abandoned.

  1. Please add data to support the claim that Nominal Catholics are “growing” (4, line 147).  If this claim is correct it needs to be analyzed and explained.  These people still self-identify as Catholics, did not become Protestants, and did not drop out of religion altogether either.

                      Point taken, see comment above.

  1. It is a strange sweeping generalization to consider the whole RCC as “priestly,” all of Pentecostalism as “prophetic”, and as indigenous/Afro-Caribbean forms as “magical” in Bourdieu’s theory (9, 415-418).  Would it not make more sense to say that the Catholic Church has priestly (Sacramental Reform), prophetic (Liberationalist), and magical (CCR, Popular, Hybrid) forms?  Likewise, one could consider the historical Protestant churches as “priestly,” neo-Pentecostalism as “prophetic,” and (classical) Pentecostalism as “magical.”

                      Point taken, I have made it explicit that a first rudimentary Bourdieuan categorization of Roman Catholicism as ‘priestly’ does not work (or is too simplifying) in Latin America when one takes closer look.

  1. The 2006 Pew report has been severely criticized for its error in conflating charismatic Catholics, Pentecostals, and neo-Pentecostals all in the same category as “renewalists” (12, line 525-530).  The 2006 report is problematic to use as a source, especially for percentages.

                      I agree. most of the numbers are not helpful due to the mixing of Neo-Pent, Pent., and Charismatic Cath. into one category. There are, however. Also, numbers for Charismatic Cath. (singles out) which are useful. Note that Hawkins explicitly creates a new religious category “Christian Pentecostalism” that includes all there types. I’m sceptical about the usefulness of that. I have added a cautionary note about all of the above in the text.

  1. I am puzzled why the CCR is listed together with “more conservative” lay movements such as the Cursillos, Legion of Mary, and Legionaries of Christ (13, 592-594).  What are the reasons for that?  The CCR is not political in purpose or practice; its leaders can be more progressive or more conservative, depending on their education, class, and home country.

                      Good comment. I agree that the CCT is not necessarily conservative (in either a political or a religious/moral/dogmatic sense). Having studied Charismatic sermons, media, rallies, I do think that it is fair to say that in a religious sense (dogma, morality) a majority tends strongly towards a conservative interpretation (however not liturgically). Regarding secular politics, I think that the observation is completely correct, the CCR is not a necessarily conservative. I have now made the above explicit in a footnote (6) in the article                 

  1. I do not agree with the characterization of the CCR as “bottom-up” (13, 611-612).  Cleary documents how various priests, bishops, and cardinal Suenens were all instrumental in supporting and spreading the CCR throughout Latin America.  After 2000-2002, the CCR was put under central clerical control, mass healings and exorcisms were cancelled, and the emphasis was shifted to lay leadership training, parish prayer groups and providing pastoral care.  This was part of a coordinated international campaign by the Vatican that severely curtailed the CCR and strongly decreased its affiliation in most countries.

                      Good point! I agree that the CCR was introduced to Lat.Am. “top-down”, and that after the year 2000, the CCR-organization is once again under clerical control. I do think however, that for a long period (70s. 89s and 90s) the movement was self-propelling and difficult to control with hundreds of lay ministries, media outlets etc. springing forth (bottom-up). Today I think one must distinguish between the CCR and Charismatic Catholicism (not attached to the CCR). The CCR has been institutionalized (top-down), but here is still plenty of independent Catholic Charismatic groups, communities, ministries etc. I have now made the above observations and distinctions explicit in the article.

  1. I think the author underestimates how far secularization has already proceeded in Latin America (just like Morello did).  Just look at the nonreligious populations of all Latin American countries in the 2020 World Christian Encyclopedia!  Guatemala has the smallest (1.4%; although Pew listed 6% in 2014), but most countries are already between 5 and 10%.  (Apart from outliers such as Cuba and Uruguay.)  Moreover, these nonreligious populations are growing strongly, especially among the younger generations (suggesting further future growth as well).  The author should explicitly mention and analyze this, especially since it fits so well with this model/approach!

                      Good point, I agree. I have created the type “Nones” (which is overlapping with ‘progressive’ laicité’) and added the numbers form the World Chr. Encycl. I have suggested further growth of the group of non-religious

  1. I think the author should explicitly address current gaps in our knowledge about (changes in) the religious landscape of Latin America and offer some concrete suggestions for future research.

                      Good point, I have addressed some of the gaps in a final sub-section of the conclusion titled “Future research”.

I am sympathetic to the author’s ambitious effort, but I am not convinced (yet) that this effort truly constitutes a new model, i.e. that it develops a new theory that is able to not only explain what is currently happening in Latin America but also to predict future developments.  I would challenge the author to explicitly state what is new about this approach and engage in some predictions for the future.  I recognize that this is a high bar to meet; a simpler solution would be to use the word approach rather than model. 

                      I have changed it to a “map”/”mapping” which is a more fitting description for what I do.

The English is very good, with only a small number of typos and grammatical errors.  I found the word choice “antidote” (14, line 645) confusing; I assume that the author really means to say “opposite” or “antithesis”?

The sentence has been corrected

If the author can address all the 9 issues I raised here in a satisfying manner I believe this article will constitute an important contribution that should definitely be published.

Round 2

Reviewer 2 Report

I am satisfied that (almost) all the changes I suggested have been followed, making the article much stronger now!  This is now an excellent article that will be cited by many scholars.

However, the author’s hurry has led to a number of new typos.  I suggest that author will go through it with a fine comb to check every word!  Unfortunately, I cannot manage do a close reading in 3 days, but these typos and errors immediately stood out to me:

p. 2, footnote 2: Pentecostals (not “Penetcostals”).

p. 4, line 152: “is type”??  Some words missing here.

p. 9, footnote 3: correct author names are Johnson and Zurlo (2020).

p. 10, line 422 and line 433: random capital letters T and I.

p. 13, footnote 7: very large huge movement.

p. 14, line 632: random capital letter T.

p. 17, line 798-799: correct author names are Johnson and Zurlo (2020).

Author Response

Dear Reviewer 2,

thank you for an encouraging review round 2 an thank you again for excellent feedback in round 1. I've now corrected the typos "Johnson/Zurlo". And I've combed through the text twice correcting minor errors.

best regards,

the  author